# Ultrasound-Guided Core Needle Biopsies of Breast Invasive Carcinoma: When One Core is Sufficient for Pathologic Diagnosis and Assessment of Hormone Receptor and HER2 Status

**DOI:** 10.3390/diagnostics9020054

**Published:** 2019-05-13

**Authors:** Hsin-Ni Li, Chuan-Han Chen

**Affiliations:** 1Department of Pathology and Laboratory Medicine, Taichung Veterans General Hospital, Taichung 40705, Taiwan; penguin19801031@gmail.com; 2Department of Radiology, Taichung Veterans General Hospital, Taichung 40705, Taiwan

**Keywords:** ultrasound (US)-guided core needle biopsy, estrogen receptor (ER), progesterone receptor (PgR), human epidermal growth factor receptor 2 (HER2), intratumoral heterogeneity

## Abstract

Ultrasound (US)-guided core needle biopsy is considered the gold standard procedure with regard to preoperative diagnosis of breast carcinomas. However, there is no clear standard for the number of cores considered to be sufficient for pathologic evaluation, including the expression of surface hormone markers and HER2 status. Images and pathologic slides demonstrating breast invasive carcinoma from a single institution were thus retrospectively reviewed over a 12 month period. The results indicated that one core is sufficient for the diagnosis of invasive carcinomas, along with a reliable assessment of hormone receptor and HER2 status in many cases. The option of applying additional cores is recommended for some cases.

## 1. Introduction

Percutaneous image-guided core needle biopsy has replaced surgical open biopsy for the analysis of breast lesions since the early 1990s [1]. Ultrasound (US)-guided core needle biopsy has become both a powerful tool and a sensitive method in the preoperative diagnosis of breast cancer [2]. However, the number of cores that would provide a reliable diagnosis remains controversial. It is presumed that a large number of cores obtained improves accuracy, while also lengthening the duration of the procedure and increasing the risk of complications. A large, one-institutional study has suggested that at least three core needle biopsies performed under a 3-dimensional ultrasound validation are sufficient in order to obtain a reliable histological diagnosis [3]. Another study employing groups of consecutive cores demonstrated that in most cases, two biopsies per lesion are enough for a clear and reliable pathologic result [4]. A recent study also revealed that performing biopsies using different-sized needles does not affect the quality of cores or the accuracy of diagnosis [5]. However, there has been no standard or guideline regarding how many samples are necessary for pathological assessment of biomarkers of breast cancer, including surface hormone receptors, the estrogen receptor (ER) and the progesterone receptor (PgR), and the human epidermal growth factor receptor 2 (HER2). The aforementioned biomarkers are routinely tested on all primary invasive breast carcinomas by immunohistochemistry (IHC) according to the recommendations by the American Society of Clinical Oncology/College of American Pathologist (ASCO/CAP) [6]. Their status is critical in guiding clinical management and predicting prognosis. Expression of these biomarkers can be highly variable within an individual tumor [7], and the clinicopathological significance and impact of intratumoral heterogeneity has been reported [8,9]. Interpretation problems and discordant results between preoperative biopsies and final excisional specimens are thus unavoidable. As a core needle biopsy offers the only material specimens available for molecular testing for neoadjuvant treatment prior to the surgical procedure, improving the quality and standardizing the quantity of the samples are important. This study aims to standardize the number of cores required to both diagnose breast carcinomas and evaluate their representation for surface hormone receptors and HER2.

## 2. Materials and Methods

### 2.1. Study Population

From January 2017 to December 2017, US-guided core biopsies of 183 breast invasive carcinomas in 169 patients from a single hospital, aged 27–85 years (54.49 ± 11.19 years), were recruited for this study. The study was carried out in a retrospective manner and informed consent from the patients was not indispensable. Ethics approval was obtained from Institutional Review Board, Taichung Veterans General Hospital, Taichung, Taiwan (ref: CE18200B, approval date: 24 July 2018).

### 2.2. Image and Procedure

Images from two-dimensional sonography applied during the procedure were reviewed to confirm the needle was in place and the necessary representative tissue obtained. All biopsies were performed by means of core needle biopsy via an automated biopsy gun fitted with 16- or 18-gauge needles.

### 2.3. Pathologic Evaluation

The pathology reports in our electronic medical records system and pathologic slides were reviewed. The numbers of cores containing diagnostic invasive foci for each lesion were recorded, in addition to the cores containing the foci of carcinoma in situ and the total tumor volume in each lesion. The expression status of ER, PgR and HER2 in each core was assessed through the use of IHC, according to the recommendations of the ASCO/CAP [6]. Immunohistochemistry was carried out with the SP1 antibody for ER, the 1E2 antibody for PR and the HER2 monoclonal antibody on a Ventana autostainer. To determine ER and PgR status, the percentage of cells with nuclear positivity was reported as a specific number, with the positive threshold of 1% being used. HER2 IHC staining was scored according to the established guidelines, with the cut-off set at more than 10% of tumor cells with circumferential membrane staining. A Ventana chromogenic Dual In-situ Hybridization (DISH) assay was used for determining the presence or absence of gene amplification if the IHC result was equivocal (score: 2+). The IHC stains were not repeated in the lesions post treatment nor in any concomitant lesions with a similar morphology in the same patient.

### 2.4. Statistics

The quantitative variables were expressed as mean ± standard deviation and qualitative variables as *n* (%). Analyses were performed using the Statistical Package for the Social Science (IBM SPSS version 22.0; International Business Machines Corp, Armonk, NY, USA).

## 3. Results

### 3.1. One Core Is Sufficient for Pathologic Diagnosis in Most Cases

The majority of patients were between 40 and 60 years of age. There were 183 invasive lesions and 51 (27.87%) of them comprised in situ foci. The obtained invasive tumor volume for each lesion varied, with the range falling between 1% and 90% of each sample. An average of 3.23 (±0.73) cores were obtained for each lesion. Of these 183 invasive lesions, only 10 (5.5%) contained cores which were not diagnosed as invasive carcinoma. The biopsies of the other 173 (94.5%) lesions demonstrated invasive foci for each core (Table 1). The total yield rate of invasive carcinoma was 97.46% (576 out of 591 cores).

### 3.2. One Core is Sufficient in Determining the Expression of Hormone Receptors and HER2 Status in Most Cases

The testing of ER, PgR and HER2 was performed in 169 lesions. In almost all cases (99.4%), it was observed that the expression of ER and PgR was consistent in each core, with the exception of two cases (Table 2, Patients No. 1 and 2). One of the cases exhibited ER nuclear positive staining in 1% of the cells in one core, and negative staining in the other three cores. The second case revealed the same picture in PgR staining, which showed positivity in one core (1% of cells), and negativity in the other three cores. In terms of HER2 testing, a greater discordance between cores was noted. Intratumoral heterogeneity was discovered in five cases (Table 2, Patients No. 3 to 7).

## 4. Discussion

It has been widely accepted in the medical community that ultrasound-guided core needle biopsy is a powerful tool in the diagnosis of breast lesions. However, there has not yet been a universal standard established for the required amount of tissue needed nor the minimum number of samples necessary for a study. Decreasing the number of cores obtained via core needle biopsy has the advantage of shortening the duration of the procedure, reducing costs, decreasing the risk of complications and easing the anxiety of patients.

The results from our study show that by applying 2-dimensional sonographic assistance to confirm correct needle placement during the procedure, almost all cores will subsequently contain diagnostic tissue (97.46%). Ten lesions which did not reach a 100% positive rate revealed certain characteristics, including small size (≤1 cm) (5/10), areas of ductal carcinoma in situ (DCIS) predominant (5/10), a cystic component (1/10) and a fibrotic texture (1/10) (Table 1). These characteristics may possibly lead to technical problems and misjudgment during the procedure. Additional biopsies are thus recommended in cases with lesions having the aforementioned features. The major issue of this recommendation is to decide whether additional cores are required before biopsies when the presence of “areas of DCIS predominant” is not known. Two approaches might be helpful in practical settings. First, when the lesions display heterogeneous echogenicity in the presence of foci of indistinct margin, which might predict different tumorous components and stromal invasion, additional biopsies are suggested. Second, instead of increasing biopsied cores, a 2-step approach is recommended in cases whose initial biopsies reveal only in situ lesions while underestimation of invasive carcinomas is highly suspected. Upstaging from DCIS detected by core needle biopsy to invasive carcinomas in the following specimens is well documented [10], and physicians can recommend repeat biopsies in case clinical suspicion of higher grade lesions or abnormal axillary lymph nodes is detected [11]. For the lesions which are small in size (≤1 cm) or hard in texture, any marginal pass or taking out of the lesions during the biopsy procedure has to be avoided. Procedures guided by more precise imaging using three-dimensional or multiplanar imaging display to define the actual position of the needle might aid in obtaining representative tissue.

The expression of ER, PgR and HER2 has been routinely tested in all samples from core needle biopsy. The advantages of using core needle biopsy for determining the status of the aforementioned biomarkers include both better therapeutic planning and prognosis prediction. Another advantage proposed is the more optimal fixation conditions compared with surgical specimens [12]. There is, however, a disadvantage for using core needle biopsy in that there is the possibility of crush artifacts, which may lead to false-positive or false-negative results [12]. In addition, intratumoral heterogeneity has been well documented within an individual breast carcinoma, in the presence of heterogeneous cell populations with different characteristics, including tumorigenicity, molecular signature, therapeutic resistance and metastatic potential [13]. Since a core needle biopsy acquires a small portion of the lesion, the overall biomarker expression may not be totally reflected. The current guidelines for the reporting of biomarkers aim to maximize patient eligibility for target therapy, and do not take into account intratumoral heterogeneity, implying that this disadvantage may be minimized. Several series have also confirmed the high concordance between preoperative core needle biopsy and resection specimens for ER and HER2 status [12].

In our study, we demonstrated that samples of one core are sufficient for determining the expression of hormone receptors in most cases, particularly for ER and PgR expression. At the low positivity threshold (1%), the heterogeneous expression may not cause a major problem in the interpretation of ER and PgR status. However, the heterogeneity of HER2 expression in different cores can lead to an inaccurate determination if only one core is obtained, which may be followed by inappropriate therapeutic decisions. For example, Patient Nos. 4 and 5 in our study were eligible for HER2 DISH testing regarding gene amplification, but may be erroneously classified as a negative group in IHC studies. Patient Nos. 6 and 7 displayed positive staining in their half samples and might therefore be placed in the wrong group due to a single core biopsy. Patient No. 3, who was classified into the positive group, might undergo HER2 DISH testing if the core with an equivocal result (score 2+) had been biopsied. Since intratumoral heterogeneity of HER2 protein expression and gene amplification has been recognized and is inevitable, repeat testing on a subsequent specimen should be considered, particularly if the tumor has histologic characteristics associated with HER2 positivity (i.e., a tumor grade of 2 or 3, a weak or negative PgR expression or an increased proliferation index) [6], or the clinical course is not as expected.

The main limitations of this study include its retrospective nature as well as its small sample size from a single center. Large-scaled and well-designed studies with increased case numbers, applying more sophisticated image-guided modalities, better radiopathologic correlation and prospective designs will be needed to validate the result.

## 5. Conclusions

In the majority of our cases, one core containing an invasive part is sufficient for the pathologic diagnosis and evaluation of predictive and prognostic markers. Representative tissue could usually be obtained through one core US-guided biopsy, except under certain circumstances, such as small size (≤1 cm), heterogeneous tumor components, and fibrotic texture. However, one core biopsy might not be applicable for all breast conditions. Thus, current established guidelines should be followed, and repeat biopsy is considered if discordance is present among clinical, radiological and pathologic evaluation.

## Figures and Tables

**Table 1 diagnostics-09-00054-t001:** Features of 183 invasive lesions biopsied.

**173 Lesions with Invasive Foci in Each Core**	**Number of Cores Biopsied**	1	2	3	4		
	**Number of Lesions**	2	25	79	67		
**10 lesions with some cores in the absence of invasive carcinomas**	**Patient No.**	**Number of Cores Biopsied**		**Cores Containing Invasive Foci**		**Size of the Lesion**	**Features of the Lesion**
	1	3		2		40 mm	DCIS preponderance
	2	4		3		9.7 mm	Status post-treatment with fibrotic change
	3	3		1		20 mm	DCIS preponderance
	4	4		1		7 mm	DCIS preponderance
	5	4		3		8 mm	
	6	4		3		10 mm	
	7	2		1		124 mm	Cystic component
	8	3		2		20 mm	
	9	4		1		20 mm	DCIS preponderance
	10	3		2		9 mm	DCIS preponderance

Abbreviation: DCIS, ductal carcinoma in situ.

**Table 2 diagnostics-09-00054-t002:** Cases of invasive breast carcinoma in the presence of heterogeneous expression of surface hormone receptors and HER2.

Biomarker	Patient No.	Number of Cores	Expression Status in Each Core	Interpretation
**ER**	1	4	1%/0/0/0	Positive
**PgR**	2	4	1%/0/0/0	Positive
**HER2**	3	3	3+/3+/2+	Positive
	4	3	1+/1+/2+	Eligible for HER2 DISH
	5	4	2+/1+/1+/1+	Eligible for HER2 DISH
	6	2	3+/1+	Positive
	7	4	3+/3+/1+/1+	Positive

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
