# Peer review of "Ultrasound-Guided Core Needle Biopsies of Breast Invasive Carcinoma: When One Core is Sufficient for Pathologic Diagnosis and Assessment of Hormone Receptor and HER2 Status"

_diagnostics, 2019, doi:10.3390/diagnostics9020054_

Round 1

Reviewer 1 Report

In this article, a retrospective study is carried out on the US-guided core biopsies of 183 invasive breast carcinomas. The study is retrospective, so it is subject to some limitations.

The introduction is adequate to present the problem addressed in the article.

The Material and methods section is brief but correct. Authors should mention that the biopsies studied come from a single hospital. In "statistics" it should be described that quantitative variables are expressed as mean ± standard deviation and qualitative variables as n (%).

The results section is concise and clear. Table 1 fails to clarify the meaning of the acronym "DCIS."

In the discussion, authors should comment more extensively on the limitations of this retrospective study.  The authors can not make the recommendation to perform a single core biopsy based solely on this study. It should be noted that this study has been carried out in a single center and retrospective. Thus, the findings reported in this article must be confirmed by well designed and prospective studies, comparing different diagnostic techniques and procedures.

Among the characteristics cited by the authors that could lead to additional biopsies is the presence of "areas of ductal carcinoma in situ predominant (5/10)". The problem is to know this fact before the biopsy containing invasive foci to decide whether to perform additional biopsies. This problem should be discussed.

In the conclusions, authors should cite again the circumstances that would advise the performance of additional cores, as well as advise new studies to confirm these findings.

Author Response

Response to Reviewer 1 Comments

Point 1: In this article, a retrospective study is carried out on the US-guided core biopsies of 183 invasive breast carcinomas. The study is retrospective, so it is subject to some limitations.

The introduction is adequate to present the problem addressed in the article.

The Material and methods section is brief but correct. Authors should mention that the biopsies studied come from a single hospital. In "statistics" it should be described that quantitative variables are expressed as mean ± standard deviation and qualitative variables as n (%).

The results section is concise and clear. Table 1 fails to clarify the meaning of the acronym "DCIS."

Response 1: Many thanks for your kindly comments. In our revised manuscript, we mention that the study come from a single institution and also describe the expression of statistics as you recommend in the Material and methods section. In addition, the full form of “DCIS”, ductal carcinoma in situ, has been added at the bottom of Table 1.

Point 2: In the discussion, authors should comment more extensively on the limitations of this retrospective study.  The authors can not make the recommendation to perform a single core biopsy based solely on this study. It should be noted that this study has been carried out in a single center and retrospective. Thus, the findings reported in this article must be confirmed by well designed and prospective studies, comparing different diagnostic techniques and procedures.

Among the characteristics cited by the authors that could lead to additional biopsies is the presence of "areas of ductal carcinoma in situ predominant (5/10)". The problem is to know this fact before the biopsy containing invasive foci to decide whether to perform additional biopsies. This problem should be discussed.

Response 2: We are very glad to have your comments and suggestions to make the study more solid. We know the retrospective study with relatively small case number can not make strong recommendations to perform a single core biopsy in every case. So a prospective study using consecutive core biopsies to see if the first core can get enough tissue for diagnosis and biomarker assessment is ongoing. Although the data haven’t well analysed, in our experience, by precise image guidance, most cases can have accurate pathologic evaluation in their first core. The finding is exciting. When we explain the procedure to the patients who are really nervous about their breast lesions, we are more confident to tell them how many cores would be enough. The study only focuses on ultrasound-guided core needle biopsy and we hope that in the future, even mammography-guided biopsy can have standardized core numbers.

In our revised version, we address two different approaches to solve the problem you mentioned: “areas of ductal carcinoma in situ predominant” is unknown before biopsies. First, we can evaluate by ultrasound whether heterogenous tumor component is present with indistinct border, which suggests different contents of the tumor and predicts invasive foci. When heterogeneous components are present in a single tumor, additional biopsy might be considered. Second, repeat biopsy is suggested if underestimation is highly suspected clinically. Of course, the group requiring repeat biopsy has to be minimized.  

Point 3:

In the conclusions, authors should cite again the circumstances that would advise the performance of additional cores, as well as advise new studies to confirm these findings.

Response 3: Again, many thanks for your helpful advice. We have added the comments in the Conclusion section.

Reviewer 2 Report

The authors present a study addressing Ultrasound-guided core needle biopsies of breast invasive carcinoma. The authors state that one core is enough to diagnose and to assess hormone receptor status. The paper is well written and the results are sound but the conclusions are not supported by the results. 5.5 % of the lesions contained cores that were not diagnosed as invasive carcinoma, which is considerable.  In terms of HER2 testing the authors identified discordance between cores but then conclude that 1 core is sufficient. Also, introduction should include more recent literature.

Author Response

Response to Reviewer 2 Comments

Point 1: The authors present a study addressing Ultrasound-guided core needle biopsies of breast invasive carcinoma. The authors state that one core is enough to diagnose and to assess hormone receptor status. The paper is well written and the results are sound but the conclusions are not supported by the results. 5.5 % of the lesions contained cores that were not diagnosed as invasive carcinoma, which is considerable. 

Response 1: Many thanks for your kindly comments. In our revised manuscript, we address more about how to decrease the percentage of failing in getting invasive parts. As we mentioned in the initial manuscript, some characteristics might affect yield rates, such as DCIS predominant, small sized lesions and fibrotic texture of the lesions. By using more precise imaging techniques (3D- or Multiplanar display), we can make sure that the needle is at the right position during the procedure to get tissue. In addition, if the tumor shows heterogeneous components, additional cores obtained shall be advised. We also modify the conclusion that one core is enough in the majority of cases. Of course, exceptional cases exist.

Point 2: In terms of HER2 testing the authors identified discordance between cores but then conclude that 1 core is sufficient.

Response 2: We admit that there is a larger discordance between cores in HER2 testing compared with that of hormone receptors. However, heterogeneous expression of HER2 protein/gene is found in a subgroup of breast carcinoma in a spatial or temporal manner, and that is why American Society of Clinical Oncology/ College of American Pathologist  (ASCO/CAP) suggests repeat biopsy and testing of this marker if clinical course or therapeutic response is not as expected. We address this issue in our discussion section and hope to have a larger-scaled and well-designed study to confirm our results.

Point 3: Also, introduction should include more recent literature.

Response 3: Many thanks for your helpful advice. We did have less statement on biomarkers in the initial manuscript. In our revised manuscript, we have added more recent literature on breast biomarkers and intratumoral heterogeneity.

Round 2

Reviewer 1 Report

The authors have responded to all the comments made about their manuscript. They have also made the required modifications to the text. Therefore I believe that the paper can be published in its present form.

Author Response

Response to Reviewer 1 Comments

Point 1: The authors have responded to all the comments made about their manuscript. They have also made the required modifications to the text. Therefore I believe that the paper can be published in its present form.

Response 1: Many thanks for your kindly comments. We do appreciate your help and valuable suggestions.

Reviewer 2 Report

One core is clearly not enough for breast biopsy as it does not include the majority of cases. Is dangerous to publish results that go against the established guidelines without not enough samples. The authors should change the conclusions according to the results, stating that the guidelines should be followed to avoid errors of diagnose in the cases where one core is not enough.  

Author Response

Response to Reviewer 2 Comments

Point 1: One core is clearly not enough for breast biopsy as it does not include the majority of cases. Is dangerous to publish results that go against the established guidelines without not enough samples. The authors should change the conclusions according to the results, stating that the guidelines should be followed to avoid errors of diagnose in the cases where one core is not enough.  

Response 1: Many thanks for your valuable suggestions. We have made modification of the conclusion section, stating that one core biopsy might not be applicable for all breast lesions. Thus, current established guidelines should be followed and additional biopsies or repeat biopsy should be applied under certain circumstances.

This manuscript is a resubmission of an earlier submission. The following is a list of the peer review reports and author responses from that submission.